# Intervention Generalization: A View from Factor Graph Models

**Gecia Bravo-Hermsdorff**[*]  **David S. Watson**[†]  **Jialin Yu**[*]  **Jakob Zeitler**[‡]  **Ricardo Silva**[*]

## Abstract

One of the goals of causal inference is to generalize from past experiments and observational data to novel conditions. While it is in principle possible to eventually learn a mapping from a novel experimental condition to an outcome of interest, provided a sufficient variety of experiments is available in the training data, coping with a large combinatorial space of possible interventions is hard. Under a typical sparse experimental design, this mapping is ill-posed without relying on heavy regularization or prior distributions. Such assumptions may or may not be reliable, and can be hard to defend or test. In this paper, we take a close look at how to warrant a leap from past experiments to novel conditions based on minimal assumptions about the factorization of the distribution of the manipulated system, communicated in the well-understood language of factor graph models. A postulated *interventional factor model* (IFM) may not always be informative, but it conveniently abstracts away a need for explicitly modeling unmeasured confounding and feedback mechanisms, leading to directly testable claims. Given an IFM and datasets from a collection of experimental regimes, we derive conditions for identifiability of the expected outcomes of new regimes never observed in these training data. We implement our framework using several efficient algorithms, and apply them on a range of semi-synthetic experiments.

## 1 Introduction

Causal inference is a fundamental problem in many sciences, such as clinical medicine [8, 73, 69] and molecular biology [66, 41, 29]. For example, causal inference can be used to identify the effects of chemical compounds on cell types [73] or determine the underlying mechanisms of disease [52].

One particular challenge in causal inference is *generalization* — the ability to extrapolate knowledge gained from past experiments and observational data to previously unseen scenarios. Consider a laboratory that has performed several gene knockouts and recorded subsequent outcomes. Do they have sufficient information to predict how the system will behave under some new combination(s) of knockouts? Conducting all possible experiments in this setting would be prohibitively expensive and time consuming. A supervised learning method could, in principle, map a vector representation of the design to outcome variables of interest. However, past experimental conditions may be too sparsely distributed in the set of all possible assignments, and such a direct supervised mapping would require leaps of faith about how assignment decisions interact with the outcome, even if under the guise of formal assumptions such as linearity.

In this paper, we propose a novel approach to the task of *intervention generalization*, i.e., predicting the effect of unseen treatment regimes. We rely on little more than a postulated factorization of the

---

[*]Department of Statistical Science, University College London.
[†]Department of Informatics, King's College London.
[‡]Department of Computer Science, University College London.

37th Conference on Neural Information Processing Systems (NeurIPS 2023).

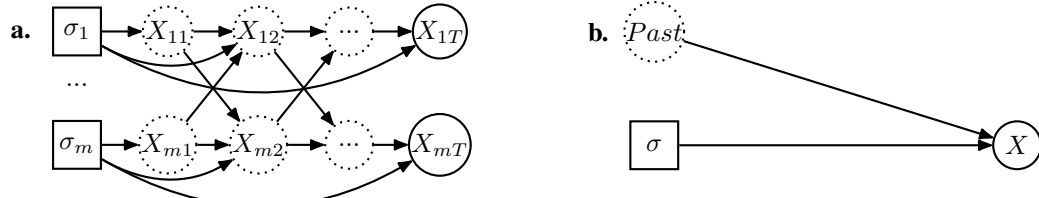

Figure 1: **(a)** A dynamic process where only the final step is (simultaneously) observed. Intervention variables are represented as white squares, random variables as circles, and dashed lines indicate hidden variables. **(b)** A bird's eye unstructured view of the sampling process, where an intervention vector dictates $\sigma$ the distribution obtained at an agreed-upon time frame $T$.

distribution describing how intervention variables $\sigma$ interact with a random process $X$, and a possible target outcome $Y$.

Related setups appear in the causal modeling literature on soft interventions [18], including its use in methods such as causal bandits [45] and causal Bayesian optimization [4] (see Section 6). However, these methods rely on directed acyclic graphs (DAGs), which may be hard to justify in many applications. Moreover, in the realistic situation where the target of a real-world intervention is a *set* of variables [26] and not only the "child" variable within a DAG factor, the parent-child distinction is blurred and previous identification results do not apply. Without claiming that our proposal is appropriate for every application, we suggest the following take on intervention generalization: *model a causal structure as a set of soft constraints, together with their putative (arbitrary but local) modifications by external actions*. Our *interventional factor model* (IFM) leads to provable intervention generalization via a factor graph decomposition which, when informative, can be tested without further assumptions beyond basic relations of conditional independence. IFMs are fully agnostic with regards to cycles or hidden variables, in the spirit of Dawid [23]'s decision-theoretic approach to causal inference, where the key ingredient boils down to statements of conditional independence among random and intervention variables.

Our primary contributions are as follows. (1) We introduce the *interventional factor model* (IFM), which aims to solve intervention generalization problems using only claims about which interventions interact with which observable random variables. (2) We establish necessary and sufficient conditions for the identifiability of treatment effects within the IFM framework. (3) We adapt existing results from conformal inference to our setting, providing distribution-free predictive intervals for identifiable causal effects with guaranteed finite sample coverage. (4) We implement our model using efficient algorithms, and apply them to a range of semi-synthetic experiments.

## 2 Problem Statement

**Data assumptions.** Fig. 1**(a)** illustrates a generative process common to many applications and particularly suitable to our framework: a perturbation, here represented by a set of interventional variables $\sigma$, is applied and a dynamic feedback loop takes place. Often in these applications (e.g, experiments in cell biology [66] and social science [59]), the sampling of this process is highly restrictive, and the time-resolution may boil down to a single *snapshot*, as illustrated by Fig. 1**(b)**. We assume that data is given as samples of a random vector $X$ collected cross-sectionally at a well-defined time-point (not necessarily at equilibrium) under a well-defined intervention vector $\sigma$. The importance of establishing a clear sampling time in the context of graphical causal models is discussed by [21]. The process illustrated in Fig. 1**(a)** may suggest no particular Markovian structure at the time the snapshot is taken. However, in practice, it is possible to model the black-box sampling process represented in Fig. 1**(b)** in terms of an *energy function* representing soft constraints [48]. These could be the result of particular equilibrium processes [47], including deterministic differential equations [40], or empirically-verifiable approximations [59].

*The role of a causal model is to describe how intervention $\sigma$ locally changes the energy function.* In the context of causal DAG models, sometimes this takes the guise of "soft interventions" and other variations that can be called "interventions on structure" [43, 55] or edge interventions [70]. Changes of structural coefficients in possibly cyclic models have also been considered [37]. The model family we propose takes this to the most abstract level, modeling energy functions via recombinations of

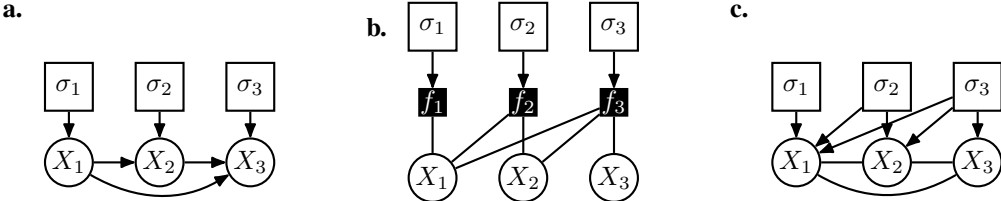

Figure 2: Examples of distinct causal graphical models, expressing different factorization assumptions. Random variables are represented as circles, intervention variables as white squares, and factors as black squares. **(a)** A directed acyclic graph (DAG) with explicit intervention variables. **(b)** The corresponding interventional factor model (IFM) for **(a)**. **(c)** A Markov random field (MRF) with interventional variables, thus, forming a chain graph.

local interventional effects without acyclicity constraints, as such constraints should not be taken for granted [22] and are at best a crude approximation for some problems at hand (e.g., [66]).

**Background and notation.** The notion of an *intervention variable* in causal inference encodes an action that modifies the distribution of a system of random variables. This notion is sometimes brought up explicitly in graphical formulations of causal models [60, 72, 23]. To formalize it, let $\sigma$ denote a vector of intervention variables (also known as *regime indicators*), with each $\sigma_i$ taking values in a finite set $\{0, 1, 2, \ldots, \aleph_i - 1\}$. A fully specified vector of intervention variables characterizes a *regime* or an *environment* (we use these terms interchangeably). Graphically, we represent intervention variables using white squares and random variables using circles (Fig. 2). We use subscripts to index individual (random or interventional) variables and superscripts to index regimes. For instance, $X_i^j$ denotes the random variable $X_i$ under regime $\sigma^j$, while $\sigma_i^j$ denotes the $i$th intervention variable of the $j$th environment. (Note that the order of the variables is arbitrary.) We occasionally require parenthetical superscripts to index samples, so $x_i^{j(k)}$ denotes the $k$th sample of $X_i$ under regime $\sigma^j$.

For causal DAGs, Pearl's *do* operator [60] denotes whether a particular random variable $X_i$ is manipulated to have a given value (regardless of the values of its parents). As an example, consider a binary variable $X_i \in \{True, False\}$. We can use a (categorical) intervention variable $\sigma_i$ to index the two "do-interventional" regimes: $\sigma_i = 1$ denotes "$do(X_i = True)$", and $\sigma_i = 2$ denotes "$do(X_i = False)$." We reserve $\sigma_i = 0$ to denote the choice of "no manipulation", also known as the "observational" regime. However, one need not think of this $\sigma_i = 0$ setting as fundamentally different than the others; indeed, it is convenient to treat all regimes as choices of data generating process.[4] Moreover, there is no need for intervention variables to correspond to deterministic settings of random variables; they may in principle describe any well-defined change in distribution, such as stochastic or conditional interventions [23, 17]. As a stochastic example, $\sigma_i = 3$ could correspond to randomly choosing $do(X_i = True)$ 30% of the time and $do(X_i = False)$ 70% of the time. As a conditional example, $\sigma_i = 4$ could mean "if parents of $X_i$ satisfy a given condition, $do(X_i = True)$, otherwise do nothing". When $X_i$ can take more values, the options for interventional regimes become even more varied. The main point to remember is that $\sigma_i = 1, \sigma_i = 2, \ldots$ should be treated as distinct categorical options, and we reserve $\sigma_i = 0$ to denote the "observational" case of "no manipulation".

As discussed in the previous section, the probability density/mass function $p(x; \sigma)$ can be the result of a feedback process that does not naturally fit a DAG representation. Indeed, a growing literature in causal inference carefully considers how DAGs may give rise to equilibrium distributions (e.g., [47, 12, 11]), or marginals of continuous-time processes (e.g., [57]). However, they come with considerable added complexity of assumptions to ensure identifiability. In this work, we abstract away all low-level details about how an equilibrium distribution comes to be, and instead require solely a model for how a distribution $p(x)$ factorizes as a function of $\sigma$. These assumptions are naturally formulated as a factor graph model [44] augmented with intervention variables, which we call an *interventional factor model (IFM)*.

---

[4]In general, there can be infinitely many choices for interventional settings. However, for the identifiability results in the next section, we cover the finite case only, as it removes the need for smoothness assumptions on the effect of intervention levels.

**Problem statement.** We are given a space $\Sigma$ of possible values for an intervention vector $\sigma$ of dimension $d$, making $|\Sigma| \leq \prod_{i=1}^{d} \aleph_i$. For a range of training regimes $\Sigma_{\text{train}} \subset \Sigma$, we are given collection of datasets $\mathcal{D}^1, \mathcal{D}^2, \ldots, \mathcal{D}^t$, with dataset $\mathcal{D}^j$ collected under environment/regime $\sigma^j \in \Sigma_{\text{train}}$. The goal is to learn $p(x; \sigma^\star)$, and $\mu(\sigma^\star) := \mathbb{E}[Y; \sigma^\star]$, for all test regimes $\sigma^\star \in \Sigma_{\text{test}} = \Sigma \backslash \Sigma_{\text{train}}$.

In each training dataset, we measure a sample of post-treatment i.i.d. draws of some $m$-dimensional random vector $X$ (possibly with $m \neq d$, as there is no reason to always assume a one-to-one mapping between intervention and random variables), and optionally an extra outcome variable $Y$. The data generating process $p(x; \sigma^j)$ is unknown. We assume $Y \perp\!\!\!\perp \sigma \mid X$ for simplicity[5], which holds automatically in cases where $Y$ is a known deterministic summary of $X$. We are also given a factorization of $p(x; \sigma)$,

$$p(x; \sigma) \propto \prod_{k=1}^{l} f_k(x_{S_k}; \sigma_{F_k}), \ \forall \sigma \in \Sigma, \tag{1}$$

where $S_k \subseteq [m]$ and $F_k \subseteq [d]$ define the causal structure. The (positive) functions $f_k(\cdot; \cdot)$ are unknown. The model $p(y \mid x)$ can also be unknown, depending on the problem.

As illustrated in Figs. 2**(a-b)**, such an intervention factor model (IFM) can also encode (a relaxation of) the structural constraints implied by DAG assumptions. In particular, note the one-to-one correspondence between the DAG factorization in Fig. 2**(a)** and the factors in the IFM in Fig. 2**(b)**: $f_1(x_1; \sigma_1) := p(x_1; \sigma_1)$, $f_2(x_1, x_2; \sigma_2) := p(x_2 \mid x_1; \sigma_2)$, and $f_3(x_1, x_2, x_3; \sigma_3) := p(x_3 \mid x_1, x_2; \sigma_3)$. This explicitly represents that the conditional distribution of $x_1$ given all other variables fully factorizes in $\sigma$: i.e., $p(x_1 \mid x_2, x_3; \sigma) \propto f_1(x_1; \sigma_1) f_2(x_1, x_2; \sigma_2) f_3(x_1, x_2, x_3; \sigma_3)$. Such factorization is not enforced by usual parameterizations of Markov random fields (i.e., graphical models with only undirected edges) or chain graphs (graphical models with acyclic directed edges and undirected edges) [25] such as the example shown in Fig. 2**(c)**.

**Scope and limitations.** The factorization in Eq. (1) may come from different sources, e.g., from knowledge about physical connections (it is typically the case that one is able to postulate which variables are directly or only indirectly affected by an intervention), or as the result of structure learning methods (e.g., [1]). For structure learning, faithfulness-like assumptions [72] are required, as conditional independencies discovered under configurations $\Sigma_{\text{train}}$ can only be extrapolated to $\Sigma_{\text{test}}$ by assuming that independencies observed over particular values of $\sigma$ can be generalized across all regimes. We do not commit to any particular structure learning technique, and refer to the literature on eliciting and learning graphical structure for a variety of methods [42]. In Appendix A, we provide a general guide on structure elicitation and learning, and a primer on causal modeling and reasoning based solely on abstract conditional independence statements, without a priori commitment to a particular family of graphical models [23].

Even then, the structural knowledge expressed by the factorization in Eq. (1) may be uninformative. Depending on the nature of $\Sigma_{\text{train}}$ and $\Sigma_{\text{test}}$, it may be the case that we cannot generalize from training to test environments, and $p(x; \sigma^\star)$ is unidentifiable for all $\sigma^\star \in \Sigma_{\text{test}}$. However, all methods for causal inference rely on a trade-off between assumptions and informativeness. For example, unmeasured confounding may imply no independence constraints, but modeling unmeasured confounding is challenging, even more so for equilibrium data without observable dynamics. If we can get away with pure factorization constraints implied by an array of experimental conditions and domain knowledge, we should embrace this opportunity. This is what is done, for instance, in the literature on causal bandits and causal Bayesian optimization [45, 49, 4, 75], which leverage similar assumptions to decide what to do next. However, we are *not* proposing a method for bandits, Bayesian optimization, or active learning. The task of estimating $p(x; \sigma^\star)$ and $\mu(\sigma^\star)$ for a novel regime is relevant in itself.

## 3 Interventional Factor Model: Identification

We define our *identification problem* as follows: given the *population* distributions $\mathbb{P}(\Sigma_{\text{train}}) := \{p(x; \sigma^1), p(x; \sigma^2), \ldots, p(x; \sigma^t)\}$ for the training regimes $\sigma^j \in \Sigma_{\text{train}}$, and knowledge of the factorization assumptions as given by Eq. (1), can we identify a given $p(x; \sigma^\star)$ corresponding to some unobserved test regime $\sigma^\star \in \Sigma_{\text{test}}$?

---

[5]Otherwise, we can just define $Y := X_m$, and use the identification results in Section 3 to check whether $Y$ can be predicted from $\sigma$.

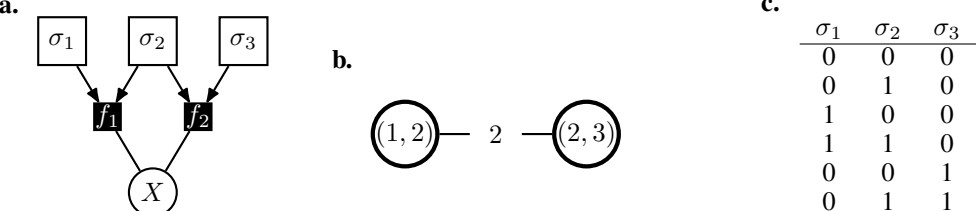

Figure 3: **(a)** An interventional factor model (IFM) with three (binary) intervention variables. **(b)** The junction tree of the $\sigma$-*graph* associated with the IFM in **(a)**: intervention variables are arranged as a hypergraph, where the hypervertices represent the sets of intervention variables that share a factor and the edge represents the overlap between the two sets of intervention variables. **(c)** A table displaying the assignments for regimes in $\Sigma_{\mathrm{train}}$. From this $\Sigma_{\mathrm{train}}$ and the assumptions in **(a)**, it is possible to generalize to the two missing regimes, i.e., $\sigma_1 = \sigma_2 = \sigma_3 = 1$, and $\sigma_1 = \sigma_3 = 1, \sigma_2 = 0$.

This identification problem is not analyzed in setups such as causal Bayesian optimization [4, 75], which build on DAGs with interventions targeting single variables. Without this analysis, it is unclear to what extent the learning might be attributed to artifacts due to the choice of prior distribution or regularization, particularly for a sparsely populated $\Sigma_{\mathrm{train}}$. As an extreme case that is not uncommon in practice, if we have binary intervention variables and never observe more than one $\sigma_i$ set to 1 within the same experiment, it is unclear why we should expect a non-linear model to provide information about pairs of assignments $\sigma_i = \sigma_j = 1$ using an off-the-shelf prior or regularizer. Although eventually a dense enough process of exploration will enrich the database of experiments, this process may be slow and less suitable to situations where the goal is not just to maximize some expected reward but to provide a more extensive picture of the dose-response relationships in a system. While the smoothness of $p(x; \sigma)$ as a function of $\sigma$ is a relevant domain-specific information, it complicates identifiability criteria without further assumptions. In what follows, we assume no smoothness conditions, meaning that for some pair $p(x; \sigma^a), p(x; \sigma^b)$, with $a \neq b$, these probability density/mass functions are allowed to be arbitrarily different. Such setting is particularly suitable for situations where interventions do take categorical levels, with limited to no magnitude information.

**Identification: preliminaries.** Before introducing the main result of this section, let us start by considering the toy example displayed in Fig. 3: Fig. 3**(a)** shows the graphical model (how $X$ could be factorized is not relevant here), and Fig. 3**(c)** shows the assignments for the training regimes $\Sigma_{\mathrm{train}}$ (all intervention variables are binary). This training set lacks the experimental assignment $\sigma_1 = \sigma_2 = \sigma_3 = 1$.[6] However, this regime is implied by the model and $\Sigma_{\mathrm{train}}$. To see this, consider the factorization $p(x; \sigma) \propto f_1(x; \sigma_1, \sigma_2) f_2(x; \sigma_2, \sigma_3)$, it implies:

$$\frac{p(x; (1,1,0))}{p(x; (0,1,0))} \propto \frac{f_1(x; (1,1))}{f_1(x; (0,1))},$$

where we used $(1, 1, 0)$ etc. to represent the $\sigma$ assignments. Because both $(1, 1, 0)$ and $(0, 1, 0)$ are in $\Sigma_{\mathrm{train}}$, the ratio is identifiable up to a multiplicative constant. Moreover, multiplying and dividing the result by $f_2(x; (1, 1))$, we get:

$$\frac{p(x; (1,1,0))}{p(x; (0,1,0))} \propto \frac{f_1(x; (1,1))}{f_1(x; (0,1))} \frac{f_2(x; (1,1))}{f_2(x; (1,1))} \propto \frac{p(x; (1,1,1))}{p(x; (0,1,1))},$$

from which, given that $(0, 1, 1)$ is also in $\Sigma_{\mathrm{train}}$, we can derive $p(x; (1, 1, 1))$.

## 3.1 Message Passing Formulation

The steps in this reasoning can be visualized in Fig. 3**(b)**. The leftmost *hypervertex* represents $(\sigma_1, \sigma_2)$ and suggests $\sigma_1$ can be isolated from $\sigma_3$. The first ratio $p(x; (1, 1, 0))/p(x; (0, 1, 0))$ considers three roles: $\sigma_1$ can be isolated (it is set to a "baseline" of 0 in the denominator) and $\sigma_3$ is yet to be considered (it is set to the baseline value of 0 in the numerator). This ratio sets the stage for the next step where $(\sigma_2, \sigma_3)$ is "absorbed" in the construction of the model evaluated at $(1, 1, 1)$. The entries in $\Sigma_{\mathrm{train}}$ were chosen so that we see all four combinations of $(\sigma_1, \sigma_2)$ and all four combinations of

---

[6]As well as the assignment $\sigma_1 = \sigma_3 = 1, \sigma_2 = 0$, which can be recovered by an analogous argument.

$(\sigma_2, \sigma_3)$, while avoiding the requirement of seeing all eight combinations of $(\sigma_1, \sigma_2, \sigma_3)$. How to generalize this idea is the challenge. Next, we present our first proposed solution.

**Definitions.** Before we proceed, we need a few definitions. First, recall that we represent the *baseline* (or "observational") regime as $\sigma_1 = \sigma_2 = \cdots = \sigma_d = 0$. This describes data captured under a default protocol, e.g., a transcriptomic study in which no genes are knocked down. Let $\Sigma^0_{[Z]}$ denote the set of all environments $\sigma^j \in \Sigma$ such that $\sigma_i^j = 0$ if $i \notin Z$. For instance, for the example in Fig. 3, a set $Z := \{2\}$ implies that $\Sigma^0_{[Z]} = \{(0,0,0),(0,1,0)\}$. Finally, let $\sigma^{[Z(\star)]}$ be the intervention vector given by $\sigma_i = \sigma_i^\star$, if $i \in Z$, and 0 otherwise. In what follows, we also use of the concepts of graph *decompositions*, *decomposable graphs* and *junction trees*, as commonly applied to graphical models [46, 20]. In Appendix B, we review these concepts.[7] An IFM $\mathcal{I}$ with intervention vertices $\sigma_1, \ldots, \sigma_d$ has an associated *$\sigma$-graph* denoted by $\mathcal{G}_{\sigma(\mathcal{I})}$, defined as an undirected graph with vertices $\sigma_1, \ldots, \sigma_d$, and where edge $\sigma_i - \sigma_j$ is present if and only if $\sigma_i$ and $\sigma_j$ are simultaneously present in at least one common factor $f_k$ in $\mathcal{I}$. For example, the $\sigma$-graph of the IFM represented in Fig. 3**(a)** is $\sigma_1 - \sigma_2 - \sigma_3$. This is a decomposable graph with vertex partition $A = \{\sigma_1\}$, $B = \{\sigma_2\}$ and $C = \{\sigma_3\}$. As this $\sigma$-graph is an undirected decomposable graph, it has a junction tree, which we depict in Fig. 3**(b)**.

**Identification: message passing formulation.** Let $\mathcal{I}$ be an IFM representing a model for the $m$-dimensional random vector $X$ under the $d$-dimensional intervention vector $\sigma \in \Sigma$. Model $\mathcal{I}$ has unknown factor parameters but a known factorization $p(x; \sigma) \propto \prod_k f_k(x_{S_k}; \sigma_{F_k})$.

**Theorem 3.1** *Assume that the $\sigma$-graph $\mathcal{G}_{\sigma(\mathcal{I})}$ is decomposable. Given a set $\Sigma_{\mathrm{train}} = \{\sigma^1, \ldots, \sigma^t\} \subseteq \Sigma$ and known distributions $p(x; \sigma^1), \ldots, p(x; \sigma^t)$, the following conditions are sufficient to identify any $p(x; \sigma^\star)$, $\sigma^\star \in \Sigma$:*

*i) all distributions indexed by $\sigma^1, \ldots, \sigma^t, \sigma^\star$ have the same support; and*

*ii) $\Sigma^0_{[F_k]} \in \Sigma_{\mathrm{train}}$ for all $F_k$ in the factorization of $\mathcal{I}$.*

*The algorithm for computing $p(x; \sigma^\star)$ works as follows. Construct a junction tree $\mathcal{T}$ for $\mathcal{I}$, choose an arbitrary vertex in $\mathcal{T}$ to be the root, and direct $\mathcal{T}$ accordingly. If $V_k$ is a hypervertex in $\mathcal{T}$, let $D_k$ be the union of all intervention variables contained in the descendants of $V_k$ in $\mathcal{T}$. Let $B_k$ be the intersection of the invervention variables contained in $V_k$ with the intervention variables contained in the parent $V_{\pi(k)}$ of $V_k$ in $\mathcal{T}$.*

*We define a* message *from a non-root vertex $V_k$ to its parent $V_{\pi(k)}$ as*

$$m_k^x := \frac{p(x; \sigma^{[D_k(\star)]})}{p(x; \sigma^{[B_k(\star)]})}, \tag{2}$$

*with the update equation*

$$p(x; \sigma^{[D_k(\star)]}) \propto p(x; \sigma^{[F_k(\star)]}) \prod_{V_{k'} \in ch(k)} m_{k'}^x, \tag{3}$$

*where the product over $ch(k)$, the children of $V_k$ in $\mathcal{T}$, is defined to be 1 if $ch(k) = \emptyset$.*

(All proofs in Appendix B.) In particular, if no factor contains more than one intervention variable, the corresponding $\sigma$-graph will be fully disconnected. This happens, for example, for an IFM derived from a DAG without hidden variables and where each intervention has one child, as in Fig. 2.

### 3.2 Algebraic Formulation

If a $\sigma$-graph is not decomposable, the usual trick of triangulation prior to clique extraction can be used [46, 20], at the cost of creating cliques which are larger than the original factors. Alternatively, and a generalization of Eqs. (2) and (3), we consider transformations of the distributions in $\mathbb{P}(\Sigma_{\mathrm{train}})$

---

[7]**Quick summary**: a junction tree is formed by turning the cliques of an undirected graph into the (hyper)vertices of the tree. Essential to the definition, a junction tree $\mathcal{T}$ must have a *running intersection property*: given an intersection $S := H_i \cap H_j$ of any two hypervertices $H_i, H_j$, all hypervertices in the unique path in $\mathcal{T}$ between $H_i$ and $H_j$ must contain $S$. This property captures the notion that satisfying local agreements ("equality of properties" of subsets of elements between two adjacent hypervertices in the tree) should imply global agreements. A decomposable graph is simply a graph whose cliques can be arranged as a junction tree.

| | $\sigma_1$ | $\sigma_2$ | $\sigma_3$ | $f_1^{00}$ | $f_1^{01}$ | $f_1^{10}$ | $f_1^{11}$ | $f_2^{00}$ | $f_2^{01}$ | $f_2^{10}$ | $f_2^{11}$ | $f_3^{00}$ | $f_3^{01}$ | $f_3^{10}$ | $f_3^{11}$ |
|---|---|---|---|---|---|---|---|---|---|---|---|---|---|---|---|
| $q_1$ | 0 | 0 | 0 | ✓ | | | | ✓ | | | | ✓ | | | |
| $q_2$ | 0 | 1 | 0 | | ✓ | | | | | ✓ | | ✓ | | | |
| $q_3$ | 1 | 0 | 0 | | | ✓ | | ✓ | | | | | | ✓ | |
| $q_4$ | 1 | 1 | 0 | | | | ✓ | | | ✓ | | | | ✓ | |
| $q_5$ | 0 | 0 | 1 | ✓ | | | | | ✓ | | | | ✓ | | |
| $q_6$ | 0 | 1 | 1 | | ✓ | | | | | | ✓ | | ✓ | | |
| $q_7$ | 1 | 0 | 1 | | | ✓ | | | ✓ | | | | | | ✓ |
| $p^\star$ | 1 | 1 | 1 | 0 | 0 | 0 | 1 | 0 | 0 | 0 | 1 | 0 | 0 | 0 | 1 |

Figure 4: Example of how to infer the target distribution $p\big(x;(1,1,1)\big)$ from the other seven possible regimes given the postulated factorization $p(x;\sigma) \propto f_1(x;\sigma_1,\sigma_2)f_2(x;\sigma_2,\sigma_3)f_3(x;\sigma_1,\sigma_3)$. We use $f_k^{ab}$ as a shorthand notation for $f_k(x;\sigma_{k_1}=a,\sigma_{k_2}=b)$, e.g., $p\big(x;(1,1,1)\big) \propto f_1^{11}f_2^{11}f_3^{11}$. For a PR-transformation $(f_1^{00}f_2^{00}f_3^{00})^{q_1} \times (f_1^{01}f_2^{10}f_3^{00})^{q_2} \times \cdots \times (f_1^{10}f_2^{01}f_3^{11})^{q_7} \; f_1^{11}f_2^{11}f_3^{11}$ up to a multiplicative constant independent of $x$, we need to find a solution satisfying: $q_1+q_5=0$, $q_2+q_6=0$, $\ldots$, $q_7=1$. (In the table, this can be seen by going through each $f_k^{ab}$ column and adding up the corresponding exponents $q_j$ when there is a tick in row $j$; the sum must agree with the corresponding entry in the final row). A solution is $q_1=q_4=q_6=q_7=1$, $q_2=q_3=q_5=-1$, corresponding to $p(x;\sigma^\star) \propto \big(p(x;\sigma^1)p(x;\sigma^4)p(x;\sigma^6)p(x;\sigma^7)\big)/\big(p(x;\sigma^2)p(x;\sigma^3)p(x;\sigma^5)\big)$.

by products and ratios. We define a *PR-transformation* of a density set $\{p(x;\sigma^1),\ldots,p(x;\sigma^t)\}$ as any formula $\prod_{i=1}^t p(x;\sigma^i)^{q_i}$, for a collection $q_1,\ldots,q_t$ of real numbers.

**Theorem 3.2** *Let $\sigma_{F_k}^v$ denote a particular value of $\sigma_{F_k}$, and let $\mathbb{D}_k$ be the domain of $\sigma_{F_k}$. Given a collection $\mathbb{P}(\Sigma_{\mathrm{train}}) := \{p(x;\sigma^1),\ldots,p(x;\sigma^t)\}$ and a postulated model factorization $p(x;\sigma) \propto \prod_{k=1}^l f_k(x_{S_k};\sigma_{F_k})$, a sufficient and almost-everywhere necessary condition for a given $p(x;\sigma^\star)$ to be identifiable by PR-transformations of $\mathbb{P}(\Sigma_{\mathrm{train}})$ is that there exists some solution to the system*

$$\forall k \in \{1,2,\ldots,l\}, \forall \sigma_{F_k}^v \in \mathbb{D}_k, \left( \sum_{i=1:\sigma_{F_k}^i=\sigma_{F_k}^v}^t q_i \right) = \mathbb{1}(\sigma_{F_k}^\star = \sigma_{F_k}^v), \qquad (4)$$

*where $\mathbb{1}(\cdot)$ is the indicator function returning 1 or 0 if its argument is true or false, respectively.*

The solution to the system gives the PR-transformation. An example is shown in Fig. 4. The main idea is that, for a fixed $x$, any factor $f_k(x_{S_k};\sigma_{F_k})$ can algebraically be interpreted as an arbitrary symbol indexed by $\sigma_{F_k}$, say "$f_k^{\sigma_{F_k}}$". This means that any density $p(x;\sigma^i)$ is (proportional to) a "squarefree" monomial $m^i$ in those symbols (i.e, products with exponents 0 or 1). We need to manipulate a set of monomials (the unnormalized densities in $\mathbb{P}(\Sigma_{\mathrm{train}})$) into another monomial $m^\star$ (the unnormalized $p(x;\sigma^\star)$). More generally, the possible set functions $g(\cdot), h(\cdot)$ such that $g(m^\star, \{m^1,\ldots,m^t\}) = h(\{m^1,\ldots,m^t\})$, with $g(\cdot)$ invertible in $m^\star$, suggest that $g(\cdot)$ and $h(\cdot)$ must be the product function (monomials are closed under products, but not other analytical manipulations), although we stop short of stating formally the conditions required for PR-transformations to be complete in the space of all possible functions of $\mathbb{P}(\Sigma_{\mathrm{train}})$. This would be a stronger claim than Theorem 3.2, which shows that Eq. (4) is almost-everywhere complete in the space of PR transformations.

The message passing scheme and the algebraic method provide complementary views, with the former giving a divide-and-conquer perspective that identifies subsystems that can be estimated without requiring changes from the baseline treatment everywhere else. The algebraic method is more general, but suggests no hierarchy of simpler problems. These results show that the factorization over $X$ is unimportant for identifiability, which may be surprising. Appendix C discusses the consequences of this finding, along with a discussion about requirements on the size of $\Sigma_{\mathrm{train}}$.

## 4 Learning Algorithms

The identification results give a license to choose any estimation method we want if identification is established — while we must make the assumption of the model factorizing according to a postulated IFM, we are not required to explicitly use a likelihood function. Theorems 3.1 and 3.2 provide ways of constructing a target distribution $p(x;\sigma^\star)$ from products of ratios of densities from $\mathbb{P}(\Sigma_{\mathrm{train}})$. This

suggests a plug-in approach for estimation: estimate products of ratios using density ratio estimation methods and multiply results. However, in practice, we found that fitting a likelihood function directly often works better than estimating the product of density ratios, even for intractable likelihoods. We now discuss three strategies for estimating some $\mathbb{E}[Y; \sigma^\star]$ of interest.

**Deep energy-based models and direct regression.** The most direct learning algorithm is to first maximize the sum of log-likelihoods $\mathcal{L}(\theta; \mathcal{D}^1, \ldots, \mathcal{D}^t) := \sum_{i=1}^{t} \sum_{j=1}^{n_i} \log p_\theta(x^{i(j)}; \sigma^i)$, where $n_i$ is the number of samples in the $i$th regime. The parameterization of the model is indexed by a vector $\theta$, which defines $p(\cdot)$ as $\log p_\theta(x; \sigma) := \sum_{k=1}^{l} \phi_{\theta_{k,\sigma_{F_k}}}(x_{S_k})$ + constant. Here, $\phi_\theta(\cdot)$ is a differentiable black-box function, which in our experiments is a MLP (aka a multilayer perceptron, or feedforward neural network). Parameter vector $\theta_{k,\sigma_{F_k}}$ is the collection of weights and biases of the MLP, a different instance for each factor $k$ *and* combination of values in $\sigma_{F_k}$. In principle, making the parameters smooth functions of $\sigma_{F_k}$ is possible, but in the interest of simplifying the presentation, we instead use a look-up table for completely independent parameters as indexed by the possible values of $\sigma_{F_k}$. Parameter set $\theta$ is the union of all $\sum_{k=1}^{l} \prod_{j \in F_k} |\aleph_j|$ MLP parameter sets. As maximizing log-likelihood is generally intractable, in our implementation we apply pseudo-likelihood with discretization of each variable $X$ in a grid. The level of discretization does not affect the number of parameters, as we take their numerical value as is, renormalizing over the pre-defined grid. Score matching [38], noise contrastive estimation [32] or other variants (e.g., [71]) could be used; our pipeline is agnostic to this choice, with further details in Appendix G. We then estimate $f_y(x) := \mathbb{E}[Y \mid x]$ using an off-the-shelf method, which in our case is another MLP. For a given $\sigma^\star$, we sample from the corresponding $p(x; \sigma^\star)$ with Gibbs sampling and average the results of the regression estimate $\widehat{f}_y(x)$ to obtain an estimate $\widehat{\mu}(\sigma^\star)$.

**Inverse probability weighting (IPW).** A more direct method is to reweight each training sample by the target distribution $p(x; \sigma^\star)$ to generate $\widehat{\mu}(\sigma^{i\star}) := \sum_{j=1}^{n_i} y^{i(j)} w^{i(j)^\star}$, where $w^{i(j)^\star}$ is the density ratio $p(x^{i(j)}; \sigma^\star)/p(x^{i(j)}; \sigma^i)$, rescaled such that $\sum_{j=1}^{n_i} w^{i(j)^\star} = 1$. There are several direct methods for density ratio estimation [53] that could be combined using the messages/product-ratios of the previous section, but we found that it was stabler to just take the density ratio of the fitted models using deep energy-based learning, just like in the previous algorithm. Once estimators $\widehat{\mu}(\sigma^{\star(1)}), \ldots, \widehat{\mu}(\sigma^{\star(t)})$ are obtained, we combine them by the usual inverse variance weighting rule, $\widehat{\mu}(\sigma^{i\star}) := \sum_{i=1}^{t} r^i \mu_{\sigma^{\star(i)}} / \sum_{i=1}^{t} r^i$, where $r^i := 1/\widehat{v}^i$, and $\widehat{v}^i := \sum_{j=1}^{n_i} (y^{i(j)})^2 (w^{i(j)^\star})^2$. This method requires neither a model for $f_y(x)$ nor Markov chain Monte Carlo. However, it may behave more erratically than the direct method described above, particularly under strong shifts in distribution.

**Covariate shift regression.** Finally, a third approach for estimating $\mu(\sigma^\star)$ is to combine models for $f_y(x)$ with density ratios, learning a customized $\widehat{f}_y(x)$ for each test regime separately. A s this is very slow and did not appear to be advantageous compared to the direct method, we defer a more complete description to Appendix D.

**Predictive Coverage.** Even when treatment effects are identifiable within the IFM framework, the uncertainty of resulting estimates can vary widely depending on the training data and learning algorithm. Building on recent work in conformal inference [78, 51, 76], we derive the following finite sample coverage guarantee for potential outcomes, which requires no extra assumptions beyond those stated above.

**Theorem 4.1 (Predictive Coverage.)** *Assume the identifiability conditions of Theorems 3.1 or 3.2 hold. Fix a target level $\alpha \in (0, 1)$, and let $\mathcal{I}_1, \mathcal{I}_2$ be a random partition of observed regimes intro training and test sets of size $n/2$. Fit a model $\widehat{\mu}$ using data from $\mathcal{I}_1$ and compute conformity scores $s^{(i)} = |y^{k(i)} - \widehat{\mu}(\sigma^k)|$ using data from $\mathcal{I}_2$. For some new test environment $\sigma^\star$, compute the normalized likelihood ratio $w^{(i)}(\sigma^\star) \propto p(x^{k(i)}; \sigma^\star)/p(x^{k(i)}; \sigma^k)$, rescaled to sum to $n/2$. Let $\widehat{\tau}(\sigma^\star)$ be the $q^{th}$ smallest value in the reweighted empirical distribution $\sum_i w^{(i)}(\sigma^\star) \cdot \delta(s^{(i)})$, where $\delta$ denotes the Dirac delta function and $q = \lceil (n/2 + 1)(1 - \alpha) \rceil$. Then for any new sample $n + 1$, we have:*

$$\mathbb{P}\big(Y^{\star(n+1)} \in \widehat{\mu}(\sigma^\star) \pm \widehat{\tau}(\sigma^\star)\big) \geq 1 - \alpha.$$

*Moreover, if weighted conformity scores have a continuous joint distribution, then the upper bound on this probability is $1 - \alpha + 1/(n/2 + 1)$.*

# 5 Experiments

We run a number of semi-synthetic experiments to evaluate the performance of the IFM approach on a range of intervention generalization tasks. In this section, we summarize our experimental set-up and main results. The code for reproducing all results and figures is available online[8]; in Appendix E, we provide a detailed description of the datasets and models; and in Appendix F we present further analysis and results.

**Datasets.** Our experiments are based on the following two biomolecular datasets: *i)* **Sachs** [66]: a cellular signaling network with 11 nodes representing phosphorylated proteins and phospho-lipids, several of which were perturbed with targeted reagents to stimulate or inhibit expression. There are 4 binary intervention variables. $\Sigma_{\text{train}}$ has 5 regimes: a *baseline*, with all $\sigma_i = 0$, and 4 "single-intervention" regimes, each with a different single $\sigma_i = 1$. $\Sigma_{\text{test}}$ consist of the remaining 11 ($= 2^4 - 5$) unseen regimes. *ii)* **DREAM** [30]: Simulated data based on a known *E. coli* regulatory sub-network with 10 nodes. There are 10 binary intervention variables, and (similarly) $\Sigma_{\text{train}}$ has a baseline regime, with all $\sigma_i = 0$, and 10 single-intervention regimes, with single choices of $\sigma_i = 1$. $\Sigma_{\text{test}}$ consists of 45 ($10 \times 9/2 = 45$) unseen regimes defined by all pairs $\sigma_i = \sigma_j = 1$.

**Oracular simulators.** The first step to test intervention generalization is to build a set of proper test beds (i.e., simulators that serve as causal effect oracles for any $\sigma \in \Sigma$), motivated by expert knowledge about the underlying system dynamics. Neither the original Sachs data nor the DREAM simulator we used provide joint intervention data required in our evaluation. Thus, for each domain, we trained two ground truth simulators: *i)* **Causal-DAG**: a DAG model following the DAG structure and data provided by the original Sachs et al. and DREAM sources (DAGs shown in Fig. 10(**a**)) and Fig. 11(**a**), respectively). Given the DAG, we fit a model where each conditional distribution is a heteroskedastic Gaussian with mean and variance parameterized by MLPs (with 10 hidden units) of the respective parents. *ii)* **Causal-IFMs**: the corresponding IFM is obtain by a direct projection of the postulated DAG factors (as done in, e.g., Fig. 2(**b**)). The likelihood is a neural energy model (Section 4) with MLPs with 15 hidden units defining potential functions. After fitting these models, we compute by Monte Carlo simulation their implied ground truths for every choice of $\sigma^*$. The outcomes $Y$ are then generated under 100 different structural equations of the form $\tanh(\lambda^\top X) + \epsilon$, with random independent normal weights $\lambda$ and $\epsilon \sim \mathcal{N}(0, v_y)$. $\lambda$ and $v_y$ are scaled such that the ground truth variance of $\lambda^\top X$, $\text{Var}(\lambda^\top X)$, is sampled uniformly at random from the interval $[0.6, 0.8]$, and set $v_y := 1 - \text{Var}(\lambda^\top X)$.

**Compared models.** We implement three variants of our proposed IFM model, corresponding to the three learning algorithms described in Section 4: *i)* **IFM1** uses deep energy-based models and direct regression; *ii)* **IFM2** uses an IPW estimator; and *iii)* **IFM3** relies on covariate shift regression. We compare these models to the following benchmarks: *i)* **Black-box**: We apply an off-the-shelf algorithm (XGBoost [14]) to learn a direct mapping from $\sigma$ to $Y$ without using $X$. This model does not exploit any structural assumptions. *ii)* **DAG**: We estimate the structural equations in an acyclic topological ordering that is consistent with the data generating process. This represents a strong baseline that exploits ground truth knowledge of the underlying causal graph. The likelihood is defined by conditional Gaussian models with mean and variances parameterized as MLPs, matching exactly one of the simulators described below.

**Results.** We evaluate model performance based on the proportional root mean squared error (pRMSE), defined as the average of the squared difference between the ground truth $Y$ and estimated $\hat{Y}$, with each entry further divided by the ground truth variance of the corresponding $Y$. Results are visualized in Fig. 5. We additionally run a series of one-sided binomial tests to determine whether models significantly outperform the black box baseline, and compare the Spearman's rank correlation [80] between expected and observed outcomes for all test regimes. Unsurprisingly, DAGs do best when the ground truth is a Causal-DAG, while IFM methods do better when the data generator is a Causal-IFM. Still, some IFMs are robust to both ground truth models, with IFM1 (the deep energy model) doing especially well on the DREAM dataset, and IFM2 (the IPW estimator) excelling on the Sachs data.

---

[8]https://github.com/rbas-ucl/intgen

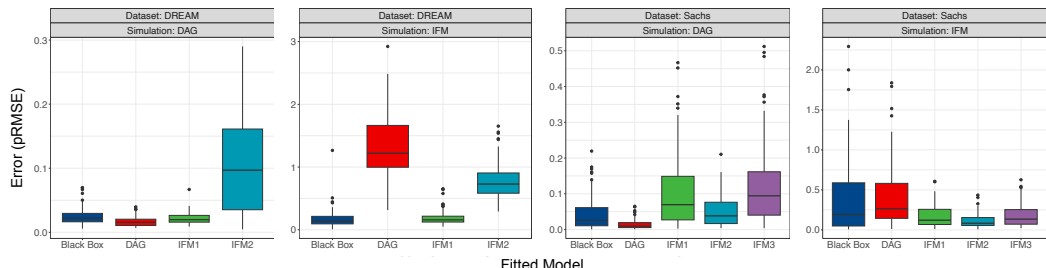

Figure 5: Experimental results on a range of intervention generalization tasks, see text for details.

In particular, for the DREAM datasets, IFM1 (scaled) errors remain stable, while the DAG has a major loss of performance when a non-DAG ground truth is presented. However, IFM2 underperform on DREAM datasets, which might be because the knockdowns regimes in those datasets induce dramatic shifts in distribution overlap among regimes. (We omit results for IFM3 (covariate shift regression) on these datasets, as the method does not converge in a reasonable time.) Though the black-box method (XGBoost algorithm and no structural assumption) sometimes struggles to extrapolate, displaying a long-tailed distribution of errors, it actually does surprisingly well in some experiments, especially on the DREAM dataset. In fact, the black-box method is essentially indistinguishable from linear regression in this benchmark (results not shown), which is not surprising given the sparsity of the training data. However, the non-additivity of the $\sigma$ effect on $Y$ is more prominent in the Sachs datasets, making it difficult to generalize without structural assumptions.

## 6 Related Work

A factor graph interpretation of Pearl's *do*-calculus framework is described in [79], but without addressing the problem of identifiability. More generally, several authors have exploit structural assumptions to predict the effects of unseen treatments. Much of this research falls under the framework of *transportability* [61, 7], where the goal is to identify causal estimands from a combination of observational and experimental data collected under different regimes. If only atomic interventions are considered, the *do*-calculus is sound and complete for this task [50]. The $\sigma$-calculus introduced by [28, 17] extends these transportability results to soft interventions [18, 19]. However, these methods are less clearly defined when an intervention affects several variables simultaneously, and while DAGs with additive errors have some identifiability [67] and estimation [75] advantages, acyclic error additivity may not be an appropriate assumption in some domains. Under further parametric assumptions, causal effects can be imputed with matrix completion techniques [73, 3] or more generic supervised learning approaches [31], but these methods often require some data to be collected for all regimes in $\Sigma_{\text{test}}$. Finally, [2] is the closest related work in terms of goals, factorizing the function space of each $\mathbb{E}[X_i; \sigma]$ directly as a function of $\sigma$.

Another strand of related research pertains to online learning settings. For instance, several works have shown that causal information can boost convergence rates in multi-armed bandit problems when dependencies are present between arms [45, 49, 24], even when these structures must themselves be adaptively learned [9]. This suggests a promising direction for future work, where identification strategies based on the IFM framework are used to prioritize the search for optimal treatments. Indeed, combining samples across multiple regimes can be an effective strategy for causal discovery, as illustrated by recent advances in invariant causal prediction [63, 33, 64, 81], and it can also help with domain adaptation and covariate shift [54, 6, 13, 15].

## 7 Conclusion

We introduced the IFM framework for solving intervention generalization tasks. Results from simulations calibrated by real-world data show that our method successfully predicts outcomes for novel treatments, providing practitioners with new methods for conducting synthetic experiments. Future work includes: *i)* integrating the approach with experimental design, Bayesian optimization and bandit learning; *ii)* variations that include pre-treatment variables and generalizations across heterogeneous subpopulations, inspired by complementary matrix factorization methods such as [2]; *iii)* tackling sequential treatments; and *iv)* diagnostics of cross-regime overlap issues [36, 58].

## Acknowledgments and Disclosure of Funding

We thank the anonymous reviewers and Mathias Drton for useful discussions. GBH was supported by the ONR grant 62909-19-1-2096, and JY was supported by the EPSRC grant EP/W024330/1. RS was partially supported by both grants. JZ was supported by UKRI grant EP/S021566/1.

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
