# OpenReview forum: "Intervention Generalization: A View from Factor Graph Models"
_NeurIPS.cc/2023/Conference — NeurIPS 2023 poster_

### Official Review · Reviewer_2Na9 · 2023-07-03

**Soundness:** 3 good
**Presentation:** 3 good
**Contribution:** 2 fair
**Rating:** 6
**Confidence:** 3

**Summary:**

This paper studies the problem of generalization in causal inference. In particular, it extends the factor graph to interventional factor graph (IFM). It shows when can such model be identified as well as proving practical algorithm for learning. The setting assumes knowing the factorization, this type of structural knowledge has its advantages and limitations as well.

**Strengths:**

The paper is well motivated with good theoretical results and empirical experiments.

**Weaknesses:**

1. This paper does not provide too much real-world example of the interventional factor model. The simulation is also semi-synthetic. I am a little concerned about the applicability of such model.
2. There seems to be a disconnect between the identifiability results and learning algorithms (see the questions section).

**Questions:**

1. Because the term junction tree is explicitly mentioned in theorem 3.1, it would be good to have a brief definition of it?
2. Can you elaborate a bit more on the algebraic formulation of this problem? This section is a bit harder to digest.
3. I am a little confused about the usefulness of theorem 3.1 and 3.2. As authors mentioned in the first paragraph of Section 4, they do not use it to product of density ratios but rather use deep energy-based model or IPW. Are the identifiability results in 3.1 and 3.2 necessary for these methods to work?

---

> ### Author Rebuttal · Authors · 2023-08-08
>
> Thanks for commenting that our paper is "well motivated with good theoretical results and empirical experiments"!
>
> **Real-world applicability:** coming from a more domain-specific theory, such structures also emerge from models of equilibrium. Consider this example that can be found in [8], Section 3.1.1, where $f$ denote differential equations at equilibrium, $X$ denotes observed random variables, $U$ denotes (mutually independent) latent variables and $I$ denotes an intervention indicator:
>
> $$f_I: X_I - U_I = 0$$
> $$f_D: U_1(X_I - X_O) = 0$$
> $$f_P: U_2(gU_3X_D - X_P) = 0$$
> $$f_O: U_4(U_5I_KX_P - X_O) = 0$$
>
> After marginalizing the $U$ variables, what we get is an IFM (this is not the whole story, though, as other non-graphical constraints may take place given the particular equations. [8] discusses $U_I$ being marginally independent of $I_K$ on top of the above). In general, energy-based models are to be interpreted as conjunctions of soft constraints, the factor graph is one implication of a system of stochastic differential equations, and interventions denote change to particular constraints. A SDE model may have *other* assumptions on top of the factorization, and parameters which carry particular meaningful interpretations, but this fits well with our claims that the IFM is a "minimalist" family of models in terms of structural assumptions - a reasonably conservative direction to follow particularly when the dynamics of many natural phenomena cannot be (currently) measured at individual level, as it's the case of much of cell biology data. [33] has further elaborations on some of these ideas.
>
> **Usefulness of Theorems 3.1 and 3.2:** it is clear that we should have been more explicit in the transition between these sections. Section 3 is about identification methods, and those don't necessarily need to translate directly into estimation methods. By analogy, think of the formulas implied by the do-calculus or G-computation, and how it's not necessarily the case they are used in estimation methods by plugging-in a corresponding estimate of a density.
>
> That is, the proofs in Section 3 are constructive, but this doesn't imply they should lead to a one-to-one correspondence with a learning algorithm (for instance, G-estimation looks very different from what a direct approach based on G-computation may suggest). As long as we know that we can get $\Sigma_{test}$ by products of ratios of densities in $\Sigma_{train}$, it doesn't matter whether we can identify each factor as long as all densities follow the common factorization that shares factors, so a purely likelihood-based approach suffices (we did try density ratio estimation methods, but they worked poorly out-of-the-box and we decided to omit further discussion on them). We will clarify this in a final manuscript by a more fleshed out bridge between Sections 3 and 4. All problems in Section 5 can be identified based on the fact that it satisfies the conditions of Theorem 1, as there is no more than one $\sigma$ per factor and $\Sigma_{train}$ spans all necessary elementwise support for each $\sigma$ variable (a scenario we mention in Section 3). We will comment on this explicitly in a revised manuscript.
>
> **Algebraic formulation:** essentially, we need to find a vector $\{q_i\}$ such that $p(x; \sigma) \propto \prod_{k = 1}^t p(x; \sigma^i)^{q_i}$, or show that there is no solution. With $x$ fixed, this is done by treating each factor $f_k(x_{S_k}, \sigma_{F_k})$ as a symbol in an algebraic system. This means that each density, up to a multiplicative constant, is a monomial in those symbols. For instance, in the example in Figure 3, we use the symbol $f_1^{00}$ to denote the factor $f_1(x; \sigma_1 = 0, \sigma_2 = 0)$, while e.g. $f_3^{10}$ denotes $f_3(x; \sigma_1 = 1, \sigma_3 = 0)$ and so on. So the monomial corresponding to $p(x; \sigma = (1, 1, 1))$ is $f_1^{11}f_2^{11}f_3^{11}$ and so on.
>
> Now, a PR transformation from the set of 7 training densities in Figure 3 (all configurations of binary $\{\sigma_1, \sigma_2, \sigma_3\}$ except for the test configuration $\sigma_1 = \sigma_2 = \sigma_3 = 1$) is of the form $$(f_1^{00}f_2^{00}f_3^{00})^{q_1} \times (f_1^{01}f_2^{10}f_3^{00})^{q_2} \times \dots \times (f_1^{10}f_2^{01}f_3^{11})^{q_7}.$$
>
> For an algebraic identity between $f_1^{11}f_2^{11}f_3^{11}$ and the above to hold (up to a multiplicative factor), it is necessary that $q_1, q_2, \dots, q_7$ are such that, in the resulting multiplication, the resulting exponents for $f_1^{11}$, $f_2^{11}$, and $f_3^{11}$ are 1, and all others are zero. For instance, $f_1^{00}$ appears in the regimes in lines 1 and 5 in the table of Figure 3. Hence, we must have $q_1 + q_5 = 0$. Symbol $f_1^{01}$ appears in lines 2 and 6, and therefore we must have $q_2 + q_6 = 0$. This goes on for all symbols (columns in the table), ending with $f_3^{11}$, which only appears in the final density, making $q_7 = 1$. This gives a system with one solution, the one shown in the caption. The admittedly ugly Eq. (4) is a compact way of describing this system of equations.
>
> **Brief definitions:** provided with the opportunity to write a camera-ready version of this paper, we will have one extra page and we will be able to move selected and summarized definitions from the appendix into the main body, including that of a junction tree.

---

> > ### Comment · Reviewer_2Na9 · 2023-08-17
> >
> > Thanks for your reply! Your responses have addressed my concerns and questions thoroughly. I will keep my score of 6 and still lean towards acceptance.

---

### Official Review · Reviewer_EjuP · 2023-07-07

**Soundness:** 3 good
**Presentation:** 2 fair
**Contribution:** 2 fair
**Rating:** 5
**Confidence:** 4

**Summary:**

This work proposes the use of factor models as a graphical causal model to generalize from past experiments. The authors introduce factor models, describe their relative merit then describe how a factorization can be derived for a given intervention, and give approaches for estimation using deep energy based modeling and weighting approaches. Experimental results are shown which provide a comparison of the performance of the proposed approach to other approaches (DAGs and black box estimation)

**Strengths:**

* This is a very interesting idea, and I generally agree with the authors' central claims around the opportunities provided by using factor models for causal inference.

* The addition of conformal inference for uncertainty intervals here is a nice, and elegant addition to the paper.

* The authors do a nice job of providing a thorough empirical evaluation of the proposed approach

**Weaknesses:**

One of the weaknesses of factor models is that it is more difficult to perform inference than in DAGs where there is a simple factorization that can be exploited. Factor models are also less immediately interpretable than DAG models. While this isn't necessary a problem in itself, it would be useful if there was a more plain discussion about the tradeoffs involved in this representation.

I also found the presentation to be a little difficult to follow. There are a number of missing discussions that would be useful to contextualize the proposed approach in the broader literature (e.g., ADMGs, segregated graphs, gated factor graphs). It's also a little unclear to me whether there is a sound and complete identification algorithm here.

**Questions:**

* Is the set of identifiable estimands comparable to other graphical causal models like ADMGs?

* For these estimation approaches (and more generally) is it possible to provide a sense of convergence/consistency of the causal parameters?

* Given the experimental results where there does not seem to be a clearly preferable approach in all settings, how should someone decide when it is appropriate/necessary to employ an IFM?

* Can you provide a discussion of the current proposal with "Causality with Gates" by Winn (AISTATS 2012). I can see that there are significant differences between these two texts, but given that they are both concerned with the use of factor graphs for causal inference I think it should be discussed.

**Limitations:**

Yes.

---

> ### Author Rebuttal · Authors · 2023-08-08
>
> Thank you for finding our idea to be "very interesting", implemented with "elegant additions" on uncertainty quantification and a "through empirical evaluation"! In what follows, we address your questions.
>
> **Trade-offs in the representation:** one of the main defining features of DAGs is the natural presence of marginal independencies between intervention variables and random variables (a random variable $X_i$ is independent of an intervention variable $\sigma_j$ when its distribution doesn't change as we choose different levels for $\sigma_j$. See [17] for further discussion about what "independence" means in the context of non-random intervention variables). That "the future doesn't cause the past" is one of the sources of this type of independence.
>
> In contrast, no marginal independencies of this type are structurally encoded in a (connected) IFM. This is more natural in a scenario where a design vector $\sigma$ is set at the initial stage in a process, and the system is runs towards some equilibrium. No element $\sigma_i$ in the intervention vector is decided based on outcomes caused by the other $\sigma_j$ in this vector. This is what happens in many datasets such as Sachs et al. Although traditionally DAGs have been applied to it, the nature of the data sampling mechanism, and the feedback generative mechanism itself, puts such a construction in dispute. References [8] and [32] have a discussion on such issues, where directed components are still used - but even there, we believe there is much to appreciate about the relative simplicity of an IFM, leading to both more tractable theoretical and practical analyses when integrating data from multiple jointly-intervened design vectors.
>
> DAGs are also natural where there is a sequential plan, i.e., instances of closed-loop control where an action is chosen based on the outcomes of previous actions. Even in this case, cross-sectional IFMs could be combined in a sequence, as an instance of chain graph modelling where a design vector is jointly decided at different stages, a next equilibrium distribution is achieved, and a consecutive round of decisions is taken based on the previous equilibrium. So, we envision that a natural extension is a chain-graph combination of directed components and IFMs. Data for that is not as easy to find as for a single-shot IFM, hence our focus here.
>
> **Soundness and completeness:** an IFM is definitely not meant to capture possible identification conditions that rely of marginal independencies, which can be exploited by DAG-based models. Likewise, the closest DAG relaxation of an IFM model will fail to capture relevant constraints (e.g., a bivariate model given by a copula $c(x_1, x_2)$ and marginals $f_1(x_1; \sigma_1)$, $f_2(x_2; \sigma_2)$). We also define a broad family of transformations (PR transformations), which we conjecture informally as encompassing all symbolic mappings from training to test regimes. Within PR transformations, we provide a sound and (measure one) complete method to show identifiability.
>
> **When to decide on the use of an IFM:** following the above discussion, it is clear that the choice of family matters. Our DAG-based simulations were designed to make the DAG alternative competitor look as good as possible, matching the exact additive-error conditionally Gaussian family. In practice, off-the-shelf goodness-of-fit techniques, even if using the baseline regime only, should provide diagnostic tools for this choice. Even plotting simulations from the fitted model $p(x|\sigma^0)$, compared against the training data, already provides strong indications of what each model can accomplish.
>
> **Estimation:** in the sense of Fisher (pointwise) consistency, learning densities with the postulated factorization, and supervised methods for mapping $X$ to $Y$, will guarantee consistency for any $\mathbb{E}[Y; \sigma]$ identifiable from the conditions in Section 3. This is true even if not all individual factors in the factorization are identifiable (up to a multiplicative constant): the constructive identification results bind training densities to test densities; having the required factorization and support is what matters.  We agree it would be interesting to have other inferential tools for the causal parameters, e.g., rates of convergence. It is not our goal to address it here but we hope this fosters further work in the community, particularly considering that the causal functionals of interest, $\mathbb{E}[Y; \sigma^\star]$, are relatively simple.
>
> **Winn (2012):** many thanks for the reference! It is an early example of how factor graphs can be used to encode dependencies between variables in a causal graph and well worth discussing in our paper. We highlight that our focus is on explicit modeling of non-atomic interventions and identifiability of out-of-distribution regimes, while Winn (2012) sets the stage on how do-calculus applied to DAGs can be translated to a type of factor graph, although it must rely also on context-specific independencies (via the "gates" variation of a factor graph model). An excellent pointer to be added to our references.

---

> > ### Comment · Reviewer_EjuP · 2023-08-21
> >
> > Thank you for your thoughtful response, and apologies for a delayed reply. I appreciate your framing on the generality of factor models versus DAGs, however it still isn't entirely clear to me when a practitioner should prefer to use the factor graph over existing frameworks, especially since identification is out of scope of this paper (contrast this to e.g., ADMGs which do admit identification). With that being said, I feel that the authors response does alleviate at least some of my concerns and I am upgrading my score to reflect this.

---

> > > ### Author Response · Authors · 2023-08-21
> > > **Thank you!**
> > >
> > > Thanks for the further consideration, we are mindful of real-world constraints and we appreciate the feedback at any stage! To be honest, we are not totally clear which missing identification results are being referred to, but no hurry. Any further detail you may be able to provide us at some point at the time decisions are released will be welcome and we will take them into full consideration.
> > >
> > > Many thanks again!

---

> ### Author Response · Authors · 2023-08-18
> **Can we help with more comments?**
>
> Thank you for all the feedback so far. Please do let us know if there is anything else for us to address in our rebuttal.
>
> All the best,
>
> the authors

---

### Official Review · Reviewer_t5cb · 2023-07-07

**Soundness:** 3 good
**Presentation:** 3 good
**Contribution:** 2 fair
**Rating:** 6
**Confidence:** 2

**Summary:**

The authors present the interventional factor model, a more general formalization used to predict the effect of treatment on an outcome in unseen regimes. It is more general than existing formalizations since it does not assume causal graphs to be directed acyclic graphs (DAGs). The authors show in this new formalization what are the conditions to ensure identifiability of treatment effects. Finally, the authors propose several methods that can estimate treatment effects and show their effectiveness with semi-synthetic experiments.

**Strengths:**

The article is well written. It proposes a really general formalization using factor graphs that is original. It encompasses identification results, several algorithms, and experiments.

**Weaknesses:**

The contribution is limited in the sense that identification results for DAGs are already well established (do-calculus and $\sigma$-calculus). This work address the more general case where the graph is not necessarily a DAG. It supposes that the graph is known, but this assumption is strong in practice since it is challenging to know these kinds of general graphs both from the expert knowledge and structure learning perspective (more than DAGs).

Real-world applications could surely help motivate the use of this formalization. The present semi-synthetic experiments are interesting, however the simple black-box baseline method is overall performing really well, undermining the use of the more involved proposed methods.

**Questions:**

Minor typos or style suggestions:
- line 39: "we submit" => "we argue" or "we claim"
- line 46: use a colon instead of a full stop
- line 59: why use the aleph symbol, could another common letter be used instead?
- line 204: "Equ." => "Eq.". It is more common and "Eq." is used for all the other references to equations.
- line 350: citation 2 is repeated

**Limitations:**

Yes the limitation have been addressed, and the societal impact is not really applicable.

---

> ### Author Rebuttal · Authors · 2023-08-08
>
> Many thanks for your comments, and by finding our article "well written" and "original"!
>
> **On identification/elicitation/learning:** we are not claiming that such a family is universally superior to DAGs, but there is no shortage of domains where energy-based formulations are more natural than directed representations, such as in data composed of snapshots of an equilibrium distribution (as in many cell biology applications). In particular, a set of equilibrium equations
> $$f_k(X_{S_k}, \sigma_{F_k}, U_k) = 0,$$
>
> where $f_k$ is a stochastic differential equation and $U_k$ is a set of hidden variables, is an IFM with hidden variables and partially deterministic factors. A full-blown SDE is likely to be challenging to specify (even more so if no dynamics are observable), but the factorization implied by it can still be used to set up an IFM. We have amended the text to motivate our method with practical use cases of this form.
>
> Regarding learning/elicitation of energy-based model structures, if an expert or structure learning algorithm is able to answer queries about Markov blankets, then that's already sufficient. A minimal procedure is as follows: start with a fully connected undirected graph among all variables. For all pairs of variables $(V_i, V_j)$, ask: "is $V_i$ conditionally independent of $V_j$ given all other variables? If yes, then remove the $(V_i, V_j)$ edge". When done, transform each clique in this graph into a factor in a IFM. That's it, but bear in mind that the definition of "conditional independence" for two regime indicators $\sigma_i$ and $\sigma_j$ is non-standard; see Section 7 of Constantinou and Dawid, "Extended conditional independence and applications in causal inference" (Annals of Statistics, 2017). For structure learning methods, we do need to bear in mind that only a limited set $\Sigma_{train}$ of interventional configurations are available, and it's possible that independencies within other ranges found in $\Sigma$ won't hold. Where data is lacking for sufficient levels of $\sigma_i$ variables, knowledge of physical/spatial structure can play a role, suggesting which direct interactions between interventions and random variables can be safely discarded or not. After all, without this knowledge, not even $do$ operations could be linked to real data.
>
> **Black-box baseline method:** The black-box algorithm did perform competitively in a fair number of experiments. This is just a matter of fact that we encountered after running them. There is no expectation that it will work in general (it's not meant to), and we did not want to perform dataset-selection of any kind.
>
> **Notation:** Thank you very much for the detailed comments about style and presentation, we will implement them (we are fond of $\aleph$ for denoting cardinality, but if this causes confusion with its more fundamental uses in set theory, we have no problem changing it).

---

> > ### Comment · Reviewer_t5cb · 2023-08-11
> >
> > I extend my gratitude to the reviewers for their comprehensive response. I tend to think that this paper should be accepted.

---

### Official Review · Reviewer_4dmo · 2023-07-07

**Soundness:** 4 excellent
**Presentation:** 3 good
**Contribution:** 3 good
**Rating:** 7
**Confidence:** 3

**Summary:**

This paper consider how to use data from past interventions to allow it to generalize to new unseen interventions. This is an important practical problem to consider, as running additional experiments is often costly/infeasible. In order to tackle this problem, the authors consider a graphical models approach, specifically they consider an interventional factor graph model (IFM). Using posit an IFM factorization of the the density $p(x; \sigma)$. Then they provide the sufficient conditions for identification, and provide a message passing algorithm to do so. Next, they discuss multiple approaches to estimate the density based on ML models, as as well via IPW methods. They also consider a covariate shift regression approach. Further, they provide a conformal inference approach to establish coverage. Lastly, they perform a number of experiments to establish the empirical efficacy of their method.

**Strengths:**

The paper tackles a very important problem, how can we use past interventions to generalize to new interventions. They propose an innovative IFM model and message passing algorithm to establish identification for this problem. Viewing this problem under this lens is an interesting one, and potentially of practical use. Further, I appreciate the authors providing examples so that its easier to understand their identification argument. I also appreciated the authors providing multiple methods for estimating the density discussed earlier. I believe providing multiple approaches is often of great practical use since no one algorithm often works in all scenarios.  In terms of empirical evaluation, I think its interesting that the authors used semi-synthetic data. I believe this is good practice, and should be followed more often.

**Weaknesses:**

Presentation: Presentation of both the regression and coverage algorithm is confusing, and seems to require a lot of additional knowledge on behalf of the reader. I do not fully understand how these algorithms proceed. For example, the deep-energy based models, lines 237-239 are very unclear. Making it clear what exactly is being fit would be very useful. More generally, being clear and rigorous regarding these things will go a long way in making the paper clearer.

Empirical Evaluation of Coverage: I did not see any simulations to this effect.

Empirical Evaluation: It is not clear to me what exactly X is in these datasets. Once again, being clear and rigorous about these details will enhance understanding, and give the reader a chance to appreciate the empirical evaluation. Further, even after reading the appendix, I do not understand how the outcomes were generated. Could the authors please clarify empirical details in the rebuttal?

Comparison to related work: The comparison to [2] is incorrect. The authors claim that a series of works including that of [2] requite data to be collected for all regimes in $\Sigma_{\text{test}}$. This is not the case. For example, the experimental design section of [2] shows that this isn't the case.

**Questions:**

I have made some suggestions in the weaknesses section. I list some other questions here.

Deep-energy based models:  Does fitting the parameter vector $\theta_{k, \sigma_{F_{k}}}$ require knowledge of the set of  variables in $F_k$? I am confused by this, and if it does require, how do we determine these variables in practice.

**Limitations:**

Yes, they have.

---

> ### Author Rebuttal · Authors · 2023-08-08
>
> Thanks for all of your feedback, for the kind words on this being a "very important problem" and "innovative" in this solution! Moreover, thanks for asking clarification questions that will definitely improve on the presentation of the paper.
>
> **Method clarification:** to fit a deep energy model, proceed as follows:
>
> 1. Each log-factor $\phi_{\theta_{k, \sigma_{F_k}}}(X_{S_k})$ is given by a MLP with a chosen number of hidden units. It is a different MLP for possible value of $\sigma_{F_k}$. For instance, in our empirical studies, there is at most one $\sigma_i$ variable associated with each factor. If $\sigma_{F_k}$ is non-empty, then we have two MLPs for factor $k$, which are $\phi_{\theta_{k, \sigma_{F_k} = 0 }}(X_{S_k})$ and $\phi_{\theta_{k, \sigma_{F_k} = 1}}(X_{S_k})$. Set $\theta_{k, \sigma_{F_k} = 0}$ are the weights and biases of that corresponding MLP/local regime.
>
> 2. We fit all parameters $\theta_{k, \sigma_{F_k}}$ by pseudo-log-likelihood. This means maximizing the sum of the univariate conditional log-likelihoods over all datasets $\mathcal D^i$ as
> $$\sum_{i = 1}^t \sum_{j = 1}^{n_t} \sum_{q = 1}^p \log p_\theta(x^{ij}_q|x^{ij}_w; \sigma^i),$$
> where $x^{ij}_q$ is the $q$-th column of the $j$-th data point of $\mathcal D^i$, and $x^{ij}_w$ are the remaining columns of the same row of the same dataset. Each entry in the sum above can be given as the negative energy function minus the corresponding log normalizing constant, which requires summing over the possible values of $X_q$ only *(apologies for not writing the equation explicitly, but after a long fight with apparent OpenReview's Markdown bugs, it seemed impossible to write the log-sum-exp and sub/superscripts we wanted to write. We will write it explicitly in the final manuscript)*.
>
> The above is then optimized by Adam, a gradient-based algorithm, and each MLP has a single hidden layer with a hyperbolic tangent activation function and a linear output layer.
>
> We fit a model for $f(X) := \mathbb{E}[Y|X]$ using a supervised learning method. In our experiments, another MLP. At test time, we evaluate any $\mathbb{E}[Y; \sigma^\star]$ by Monte Carlo, that is, we generate a large sample from the learned $p(x; \sigma)$ by Gibbs sampling, then average $f(X)$ over this sample.
>
> Simulations of coverage are very expensive (it requires generating at least dozens of datasets from the same model, and fitting a deep-energy based model to each to get a single evaluation). We have begun work on this, but experiments are ongoing and will not be ready ahead of the rebuttal period deadline. Our final manuscript will include coverage results either in the appendix or main text, if space allows.
>
> Without prior knowledge of what pseudo-likelihood means, we agree that it is tedious to get this from the Julia code provided in the supplement, and we will tweak the supplementary material to add the above. As we say in the main text, we have no reason to be attached to pseudo-likelihood, and one should feel free to use other methods for energy-based learning such as score matching, contrastive divergence, etc.
>
> **What exactly $X$ is in these dataset:** In the datasets, the intermediate variables $X$ refer to concentrations of metabolites in cells, such as lipids or, in the case of DREAM, (simulated) gene expression. The Jupyter notebook and the original references provide a description of the variables; see also Appendix D.
>
> **Outcome models:** they are artificially generated as follows for each of the 100 simulated problems in each of the two studies. Basically, for each of the 100 cases studies we artificially generate a vector $\lambda_{true}$ and simulate outcome data as $$Y = \tanh(\lambda_{true}^{\mathsf T}X) + \epsilon_y.$$
>
> The scale of $\lambda_{true}$ and the variance of $\epsilon_y$ are set so that $var(\lambda_{true}^{\mathsf T}X)$ under the baseline regime is about 1.5 to four times $var(\epsilon_y)$, with the ratio chosen uniformly at random from $[1.5, 4]$. For each regime $\sigma$, we then numerically compute the ground truth $\mathbb{E}[Y; \sigma]$ by Monte Carlo.
> For more details, we refer reviewers to the section "Synthetic Ground Truth and Training Data Generation" in the Jupyter notebook `starting_demo.ipynb`, where this process is described more explicitly without requiring much code reading.
>
> **"...knowledge of the set of variables in $F_k$...":** Yes, it does. How to get it? Similar to stages in causal discovery algorithms that start from an undirected network (like the PC algorithm), we can construct an undirected network that starts fully connected, and repeatedly ask an independence oracle (human expert or statistical test) whether two variables $V_i$ and $V_j$ are conditionally independent given all other variables. Special attention should be paid about the meaning of "conditional independence" when speaking of two intervention variables, see e.g. Section 7 of Constantinou and Dawid, "Extended conditional independence and applications in causal inference" (Annals of Statistics, 2017). This is how Figure 1c comes to be if we query an independence constraint oracle that follows the independencies entailed by the DAG in Figure 1a. The cliques in this graph (ignoring the directionality) can conservatively be translated to factors in a factor graph, as in Figure 1b. (There are more details about this in the response to T46G).
>
> **Comparison to [2]:** excellent point, we intended to say that about [23]. Approach [2] is very interesting but with widely different assumptions (and data requirements), more akin to sparse ANOVA in Fourier space if we were to give a very simplified description of it. We attempted to fit our very-sparse-design data with their code, but we did not succeed in getting meaningful stable results -- to be fair, these are not cases where [2] is meant to be used anyway. We will amend the text to clarify this point.

---

> > ### Comment · Reviewer_4dmo · 2023-08-11
> > **Response to Rebuttal**
> >
> > Thank you for the comments, and clarifications. I still believe that a score of 7 is a fair score of this paper, and I will keep it as such.

---

### Official Review · Reviewer_ybr1 · 2023-07-26

**Soundness:** 3 good
**Presentation:** 3 good
**Contribution:** 3 good
**Rating:** 5
**Confidence:** 3

**Summary:**

This paper maps from available observational and experimental datasets to unseen interventional distributions given the factorization of the joint distribution of the intervened system. They utilize an interventional factor model equipped with factor graphs to provide necessary and sufficient conditions for causal effect identifiability. Finally, they provide some practical algorithms to estimate the final outcomes.

**Strengths:**

The authors provided good examples of different concepts in the main paper and in the appendix. It's a different approach from approaches that use causal graphs but is quite interesting. The paper has a nice flow in the writing and is easy to read. The explanations and detailed examples provided in the appendix are worth appreciating, and they are quite helpful for readers.

**Weaknesses:**

Main weakness:

* Some concepts such as junction tree, hypervertex, and message passing algorithms should have been defined with short examples in the main paper since they are used in the main paper theorems.

* Multiple approaches have been described in sections 3 and 4 and but most of them were not explained with enough details and intuition.

* The main contribution seems a little unclear. The authors first discussed two approaches in section 3. Later in section 4, they mentioned that in practice using deep energy-based models works better. They suggested employing a differentiable black box to learn parameters for each factor and estimate E[Y| x]. Using a black box is not completely novel.
It appears that the approaches in section 3 are not very useful and thus the authors are proposing three more methods that work better in practice. I would request the authors to clarify the mentioned issues.

* In the experiment section, the authors completely ignore the approaches they discussed in Theorem 3.1 or Theorem 3.2. They considered the deep learning approach as the best version and compared the benchmarks with that.

Minor comments:
* Some concepts in sections 1 and 2 are used without proper definitions and examples. Readers would need to know those background knowledge beforehand to go with the flow of the paper.
* Line 119: Unmeasured confounders are used without any definitions or proper examples.
* Line 207:  Theorem 3.2 seems less intuitive. A proof sketch in the main paper would be appreciated.
* Line 227: The approach “Deep energy-based models and direct regression” should be described in more detail since this approach worked better than other approaches.
* Conformity scores are not defined in the main paper although has been used in theorem 4.1.


I have read the author's rebuttal. The authors resolved some of my concerns. But I am not confident enough to increase the scores.

**Questions:**

I would request the authors to provide explanations for the previously mentioned main weaknesses and answer the following questions:
* For the example in line 74, why is the DAG $\sigma \rightarrow X$, $\sigma \rightarrow  Y  \leftarrow X$, not an option?
* Are the authors refuting the utility of theorem 3.1 and 3.2 and adopting only the new approaches provided in section 4?
* How is this paper dealing with cycles and confounders? The authors should mention that more explicitly.

**Limitations:**

If the variables have more than two states or we have more variables, the number of interventional datasets will also be significantly high. In real-life applications, collecting these many interventional datasets are not feasible or possible. The authors should discuss this challenge in detail.

---

> ### Author Rebuttal · Authors · 2023-08-08
>
> Thank you for commenting that our paper is "quite interesting" and that it "has a nice flow in the writing and is easy to read". Much appreciated!
>
> In the following, we address your comments and questions.
>
> **"Some concepts... should have been defined... .":** We were struggling for space and we decided that concepts which are present in textbooks would be referred to without explicit definition in the main body of the text, but we fully agree that this isn't ideal. As a possible camera-ready allows for another page, we will take this opportunity to use it for modifications like that if given the opportunity.
>
> **"...details and intuition.":** We tried to present a minimal example of the junction tree approach in lines 151-159. We will flesh it out a bit more, and point more explicitly to Appendix A where a more complex example is given, assuming the opportunity to provide an accepted camera-ready version. Likewise, we will explicit write further intermediate steps into the explanation of the algebraic method in the caption of Figure 3. It can be explained as follows.
>
> Essentially, we need to find a vector $\{q_i\}$ such that $p(x; \sigma) \propto \prod_{k = 1}^t p(x; \sigma^i)^{q_i}$, or show that there is no solution. With $x$ fixed, this is done by treating each factor $f_k(x_{S_k}, \sigma_{F_k})$ as a symbol in an algebraic system. This means that each density, up to a multiplicative constant, is a monomial in those symbols.
>
> For instance, in the example in Figure 3, we use the symbol $f_1^{00}$ to denote the factor $f_1(x; \sigma_1 = 0, \sigma_2 = 0)$, while e.g. $f_3^{10}$ denotes $f_3(x; \sigma_1 = 1, \sigma_3 = 0)$ and so on. The monomial corresponding to $p(x; \sigma = (1, 1, 1))$ is $f_1^{11}f_2^{11}f_3^{11}$.
>
> Now, a PR transformation from the set of 7 training densities in Figure 3 (all configurations of binary $\{\sigma_1, \sigma_2, \sigma_3\}$ except for the test configuration $\sigma_1 = \sigma_2 = \sigma_3 = 1$) is of the form $$(f_1^{00}f_2^{00}f_3^{00})^{q_1} \times (f_1^{01}f_2^{10}f_3^{00})^{q_2} \times \dots \times (f_1^{10}f_2^{01}f_3^{11})^{q_7}.$$
>
> For an algebraic identity between $f_1^{11}f_2^{11}f_3^{11}$ and the above to hold (up to a multiplicative factor), it is necessary that $q_1, q_2, \dots, q_7$ are such that, in the resulting multiplication, the resulting exponents for $f_1^{11}$, $f_2^{11}$, and $f_3^{11}$ are 1, and all others are zero. For instance, $f_1^{00}$ appears in the regimes in lines 1 and 5 in the table of Figure 3. Hence, we must have $q_1 + q_5 = 0$. Symbol $f_1^{01}$ appears in lines 2 and 6, and therefore we must have $q_2 + q_6 = 0$. This goes on for all symbols (columns in the table), ending with $f_3^{11}$, which only appears in the $7^{th}$ training density, making $q_7 = 1$. This gives a system with one solution, the one shown in the caption. The admittedly ugly Eq. (4) is a compact way of describing this system of equations.
>
> **"It appears that the approaches in section 3 are not very useful...":**. It is clear that we should have been more explicit in the transition between these sections. Section 3 is about identification methods, and those don't necessarily need to translate directly into estimation methods. By analogy, think of the formulas implied by the do-calculus or G-computation, and how it's not necessarily the case they are used in estimation methods by plugging-in a corresponding estimate of a density.
>
> That is, the proofs in Section 3 are constructive, but this doesn't imply they should lead to a one-to-one correspondence with a learning algorithm (for instance, G-estimation looks very different from what a direct approach based on G-computation may suggest). As long as we know that we can get $\Sigma_{test}$ by products of ratios of densities in $\Sigma_{train}$, it doesn't matter whether we can identify each factor as long as all densities follow the common factorization that shares factors, so a purely likelihood-based approach suffices (we did try density ratio estimation methods, but they worked poorly out-of-the-box and we decided to omit further discussion on them). We will clarify this in a final manuscript by a more fleshed out bridge between Sections 3 and 4. All problems in Section 5 can be identified based on the fact that it satisfies the conditions of Theorem 1, as there is no more than one $\sigma$ per factor and $\Sigma_{train}$ spans all necessary elementwise support for each $\sigma$ variable (a scenario we mention in Section 3). We will comment on this explicitly in a revised manuscript.
>
> **"dealing with cycles and confounders?":** We do not model confounders/cycles explicitly. As the only information that matters is the Markov blanket, the presence of an edge may be due to confounders or cycles or "directed dependence", and for better or worse that's all treated on the same footing. For instance, if the generative model happens to be a DAG with one latent variable $U$ and edges
> $$X_1 \leftarrow U, X_1 \leftarrow \sigma_1;$$
> $$X_2 \leftarrow U, X_2 \leftarrow X_1, X_2 \leftarrow \sigma_2;$$
> $$X_3 \leftarrow X_2, X_3 \leftarrow \sigma_3,$$
>
> then one factorization of the above after marginalizing $U$ is defined by two factors, $f_1(x_1, x_2; \sigma_1, \sigma_2)$, and $f_2(x_2, x_3; \sigma_3).$ The presence of $U$ makes identification harder due to the induced interaction of $\sigma_1$ and $\sigma_2$, but the implied factorization by itself doesn't introduce structural constraints not found in the original DAG. With more confounders, we may get a fully uninformative model, see the **Scope and Limitations** section.
>
> **"Why isn't [this DAG] an option?":** It may certainly be the case that a system has edges from $\sigma$ to $Y$. We don't cover this case. A case where $Y$ may be shielded from $X$ is when $Y$ is a longer-term phenotype that is a predictable result from *which* values $X$ attain at some point in equilibrium, without requiring to know *how* ($\sigma$) those measurements came to be.

---

> > ### Comment · Reviewer_ybr1 · 2023-08-16
> >
> > Thanks to the authors for their clarifications.
> > I would like to hear the authors' opinions and explanations about the limitations I have mentioned for this paper such as the high number of interventional datasets requirement.

---

> > > ### Author Response · Authors · 2023-08-17
> > > **Number of datasets**
> > >
> > > Thanks for the opportunity to follow-up on this (we ran out of space in the original box, and we thought it would be confusing to use a global response.)
> > >
> > > When the number of values per variable is particular high, we can parameterize a potential function as a smooth function of the values of each intervention variable. The identifiability results can show whether for a particular training grid of points we can identify (say) the product space over the combination of training intervention values, and uncertainty quantification methods can be used to indicate to which extent we have information to interpolate/smooth over test treatment levels points that lie in between the training levels.
> > >
> > > (To illustrate this with an analogy to the DAG case, imagine that we had each conditional distribution for a given random variable not as set of independent regressions - one for each combination of treatment parents values - but instead as a smooth mapping such as a Gaussian process with a real-valued encoding of the treatments as input)
> > >
> > > For a large number of variables, this is a generally hard problem overall regardless of method. However, the identification problem is with respect to a test set that does not need to span all possible combinations of interventions to be useful (e.g. what to focus on may be even lower-order interactions, such as in the pairwise DREAM analysis we use as illustration), with the junction tree approach showing a divide-and-conquer structure of subproblems which can be solved without providing solutions to larger joint set of variables.
> > >
> > > Many thanks again!

---

### Official Review · Reviewer_T46G · 2023-07-27

**Soundness:** 2 fair
**Presentation:** 3 good
**Contribution:** 2 fair
**Rating:** 6
**Confidence:** 4

**Summary:**

The paper introduce Interventional Factor Models (IFM), a graphical model that encodes assumptions about a data-generating process and the interventions that can be performed on top of it. The model explicitly includes intervention variables to impose a factorization over the distributions generated by the model under any interventional regime.

Being a variable Y a function of the other observable variables X (or at least independent of the interventional regimes given X), the problem setting aims to learn E[Y] under a particular interventional regime of interest, given a subset of all possible interventional distributions and a corresponding IFM. For this task, the paper presents identification criteria and learning/estimation algorithms to find the parameter of interest.

The authors present the result of experiments on semi-synthetic data to evaluate the performance of their approach.

I have some questions related to the scope of the results in the paper. I leave them in the comments section and would like to hear from the authors. I'm willing to update my scores based on the response.

**Strengths:**

- The paper presents a novel graphical model that can be used to generalize known experimental settings to new unseen settings of interest.
- Relevant setup, definitions, and results needed to understand the result are included.
- The writing is clear enough to understand the setting and main contributions of the paper.
- The paper presents sound results for the identification of causal effects and algorithms to estimate it, based on seen experimental settings and an IFM.

**Weaknesses:**

- The introduction and contributions claim that Thm. 3.2 is necessary, in general. However, the statement talks about PR-transformations in particular. It is not clear to me if it is proven that there are no other identification routes than PR-transformations.
- It is mentioned in the paper that the IFM graph could be elicited from domain experts. This is something that sounds natural to me in the case of DAG, but not so intuitive in this case. I believe the reader could benefit from a more explicit example where variables have associated meaning and one could make sense of the assumptions.

**Questions:**

- When Thm. 3.2 says no, is it the case that there is no way to identify the causal effect of interest from the given model and distributions, even if it is not in the form of a PR-transformation?
- If not, does it mean that Thm. 3.2. is necessary only under PR-transformations?

**Limitations:**

I believe the discussion on limitations and societal impacts is sufficient.

---

> ### Author Rebuttal · Authors · 2023-08-08
>
> Thank you very much for the appreciation and excellent questions! We found the clarification questions about the generality of identifiability to be particularly useful for readers.
>
> **On Theorem 3.2:** Indeed we don't make any claims of completeness outside of the class of PR transformations. We do conjecture though that PR transformations include all "practical" ways of finding a mapping from distributions in $\Sigma_{train}$, which we can explain as follows. If we fix $x$ and think of each factor $f_k(x; \sigma)$ as a symbol indexed by $\sigma$ (as in the caption of Figure 3), then each unnormalized density in each regime is a monomial over those symbols. A symbolic mapping from the set of monomials encoded by $\Sigma_{train}$ to a target monomial of interest (i.e., a test set unnormalized density) boils down to ratios and products, as monomials are closed "only" under multiplication (we didn't elaborate that the exponents in the PR transformation will be integers, as this is not necessary to assume in order to solve it). We will clarify in the  manuscript to which extent PR transformations are general enough and adjust the abstract accordingly.
>
> **IFM elicitation:** similar to stages in causal discovery algorithms that start from an undirected network (like the PC algorithm), we can construct an undirected network that starts fully connected, and repeatedly ask an independence oracle whether two variables $V_i$ and $V_j$ are conditionally independent given all other variables. Special attention should be paid about the meaning of "conditional independence" when speaking of two intervention variables, see e.g. Section 7 of Constantinou and Dawid, "Extended conditional independence and applications in causal inference" (Annals of Statistics, 2017). This is how Figure 1c comes to be if we query an independence constraint oracle that follows the independencies entailed by the DAG in Figure 1a. The cliques in this graph (ignoring the directionality) can conservatively be translated to factors in a factor graph, as in Figure 1b.
>
> The independence constraint oracle can be substituted by independence constraint/interaction tests (including the knowledge that we know that $\sigma_i$s are regime indicators and hence their sampling distribution is chosen by design). Despite relying on interventional data, faithfulness-like assumptions still play a role e.g. pairwise independence leading to joint independence. Moreover, as we observe only a limited number of values of $\Sigma$, assumptions about the lack of contextual independencies also play a role (e.g., if $X_i$ doesn't change in distribution when $\sigma_j$ changes from 0 to 1 while keeping all other $\sigma$ variables at 0, we may want to assume that this implies that $X_i$ and $\sigma_j$ should not be in any factor, regardless of the configuration of $\sigma$).
>
> Coming from a more domain-specific theory, such structures also emerge from models of equilibrium. Consider this example that can be found in [8], Section 3.1.1, where $f$ denote differential equations at equilibrium, $X$ denotes observed random variables, $U$ denotes (mutually independent) latent variables and $I$ denotes an intervention indicator:
>
> $$f_I: X_I - U_I = 0$$
> $$f_D: U_1(X_I - X_O) = 0$$
> $$f_P: U_2(gU_3X_D - X_P) = 0$$
> $$f_O: U_4(U_5I_KX_P - X_O) = 0$$
>
> After marginalizing the $U$ variables, what we get is an IFM (this is not the whole story, though, as other non-graphical constraints may take place given the particular equations. [8] discusses $U_I$ being marginally independent of $I_K$ on top of the above). In general, energy-based models are to be interpreted as conjunctions of soft constraints, the factor graph is one implication of a system of stochastic differential equations, and interventions denote change to particular constraints. A SDE model may have *other* assumptions on top of the factorization, and parameters which carry particular meaningful interpretations, but this fits well with our claims that the IFM is a "minimalist" family of models in terms of structural assumptions - a reasonably conservative direction to follow particularly when the dynamics of many natural phenomena cannot be (currently) measured at individual level, as it's the case of much of cell biology data. [33] has further elaborations on some of these ideas.

---

> > ### Comment · Reviewer_T46G · 2023-08-17
> >
> > Thanks for addressing my questions and comments.
> >
> > I still feel that the paper needs more concrete examples in terms of connecting with real-world systems. This is especially important when introducing a new family of graphical models that comes with a particular kind of implied assumptions. This gives the reader not only a better understanding of the model, but also a sense of how could those models be elicited in practice and the kind of systems they could be more useful.
> >
> > Together with the previous point, the concrete examples could help understanding the other limitation of the results, namely, needing an outcome Y that becomes independent of the treatment regimes given the other variables. The possible impact of the results critically depends on whether interesting systems fit this assumption, and if there are advantages in using IFM+results instead of other models.
> >
> > Having said that, I'm raising my score to a 6 after the authors response.

---

### Author Rebuttal · Authors · 2023-08-08

Thank you all for such detailed and helpful reviews! Getting the time and attention of no fewer than six experts is no small privilege.

Although there is some overlap among questions, it's not too substantive. As there are many reviewers, we think it will be most convenient for them if we make our rebuttals self-contained, so there will be some degree of repetition in the replies.

Again, many thanks for your help!

---

### Decision · Program_Chairs · 2023-09-21

**Decision:**

Accept (poster)

**Comment:**

The authors propose using factor graph graphical models to build a bridge between observational and interventional distributions. Pearl does this with SCMs and CBNs. Authors argue there are several practical settings where acyclicity assumption is not warranted, and fitting the SCM framework requires strong assumptions in such cases. They suggest using a collection of "intervention nodes" to represent any possible intervention.

The core techniques in the paper are not new. But I think using factor graphs to represent different interventions, and showing that one can use this probabilistic model to extrapolate across interventional distributions is a nice modeling tradeoff, in between a causally sufficient graph where every intervention can be estimated vs. no structure between interventional distributions where nothing can be done.

My main comments are below:

- As also specified by many reviewers, authors should make a more serious effort in comparing with the DAG-based modeling. For example, DAGs and undirected graphs can model distributions that cannot be modeled by each other (under faithfulness at least). There is no discussion on what kind of distributions can be modeled by factor models vs. DAGs in a natural way. The only DAG comparison is given in Figure 1 on a non-representative DAG where the DAG imposes no conditional independence.

- There is no discussion on the number of intervention requirements. Clearly one will need more interventions than a DAG-based model, but by how much? Demonstrative examples comparing ADMGs (causal graphs with latents) by possibly pulling some examples from the transportability literature of Bareinboim and Pearl would be nice to add to showcase the tradeoff in terms of how many interventions are required to transport across the two modeling frameworks.

I am still in favor of acceptance. But I strongly encourage the authors to address the above concerns and the reviewer feedback to correctly position their paper, and ensure their contribution will be conveyed more clearly to a general audience.